# Using an Aluminum Hydroxide–Chitosan Matrix Increased the Vaccine Potential and Immune Response of Mice against Multi-Drug-Resistant *Acinetobacter baumannii*

**DOI:** 10.3390/vaccines11030669

**Published:** 2023-03-16

**Authors:** Túllio T. Deusdará, Mellanie K. C. Félix, Helio de S. Brito, Edson W. S. Cangussu, Wellington de S. Moura, Benedito Albuquerque, Marcos G. Silva, Gil R. dos Santos, Paula B. de Morais, Elizangela F. da Silva, Yury O. Chaves, Luis Andre M. Mariúba, Paulo A. Nogueira, Spartaco Astolfi-Filho, Enedina N. Assunção, Sabrina Epiphanio, Claudio R. F. Marinho, Igor V. Brandi, Kelvinson F. Viana, Eugenio E. Oliveira, Alex Sander R. Cangussu

**Affiliations:** 1Graduate Program for Biodiversity and Biotechnology of Legal Amazon, Federal University of Tocantins, Palmas 77001-090, TO, Brazil; 2Graduate Program in Biotechnology, Federal University of Tocantins, Gurupi 77425-000, TO, Brazil; 3Instituto Leônidas e Maria Deane, Oswaldo Cruz Foundation-Fiocruz Amazônia, Manaus 69057-070, AM, Brazil; 4Laboratory of DNA Technology, Biotechnology Department, Multidisciplinary Support Center, Federal University of Amazonas, Manaus 69080-900, AM, Brazil; 5Department of Immunology, Biomedical Science Institute, University of São Paulo (USP), São Paulo 05508-060, SP, Brazil; 6Institute of Agricultural Sciences, Federal University of Minas Gerais, Montes Claros 39400-310, MG, Brazil; 7Department of Biotchnology, State University of Montes Claros, Montes Claros 39401-089, MG, Brazil; 8Interdisciplinary Center for Life Sciences and Nature, Federal University of Latin American Integration (UNILA), Foz do Iguaçu 85866-000, PR, Brazil; 9Departamento de Entomologia, Universidade Federal de Viçosa, Viçosa 36570-900, MG, Brazil

**Keywords:** multi-drug-resistant, *Acinetobacter baumannii*, immunosuppression model, inactivated whole-cell vaccine

## Abstract

*Acinetobacter baumannii* is a Gram-negative, immobile, aerobic nosocomial opportunistic coccobacillus that causes pneumonia, septicemia, and urinary tract infections in immunosuppressed patients. There are no commercially available alternative antimicrobials, and multi-drug resistance is an urgent concern that requires emergency measures and new therapeutic strategies. This study evaluated a multi-drug-resistant *A. baumannii* whole-cell vaccine, inactivated and adsorbed on an aluminum hydroxide–chitosan (mAhC) matrix, in an *A. baumannii* sepsis model in immunosuppressed mice by cyclophosphamide (CY). CY-treated mice were divided into immunized, non-immunized, and adjuvant-inoculated groups. Three vaccine doses were given at 0D, 14D, and 28D, followed by a lethal dose of 4.0 × 10^8^ CFU/mL of *A. baumannii*. Immunized CY-treated mice underwent a significant humoral response, with the highest IgG levels and a higher survival rate (85%); this differed from the non-immunized CY-treated mice, none of whom survived (*p* < 0.001), and from the adjuvant group, with 45% survival (*p* < 0.05). Histological data revealed the evident expansion of white spleen pulp from immunized CY-treated mice, whereas, in non-immunized and adjuvanted CY-treated mice, there was more significant organ tissue damage. Our results confirmed the proof-of-concept of the immune response and vaccine protection in a sepsis model in CY-treated mice, contributing to the advancement of new alternatives for protection against *A. baumannii* infections.

## 1. Introduction

*Acinetobacter baumannii* is an opportunistic Gram-negative, immobile, aerobic coccobacillus that causes pneumonia, septicemia, urinary tract infections, and meningitis, especially in immunocompromised patients who are hospitalized for long periods or who undergo invasive procedures with the prior use of antimicrobials [1,2,3,4,5]. *A. baumannii* is an emerging multi-drug organism that exists as resistant (i.e., resistant to three or more classes of antimicrobials), extensively drug-resistant (i.e., resistant to all classes of antimicrobials but one or two), or pan-drug-resistant (i.e., resistant to all existing antimicrobial classes) [6,7]. Multi-drug resistance is a multifactorial process; it occurs through the production of the chromosomal AmpC beta-lactamase enzyme and intrinsic oxacillinase, the loss of porin expression, the overexpression of active expulsion systems, changes in outer membrane permeability, antimicrobial efflux, and changes in drug binding sites [8,9]. It resists carbapenems by acquiring genes encoding β-lactamase, belonging to the bla OXA-gene cluster 23, which includes OXA-27, OXA-49, and OXA-73; this alteration gives rise to drug permeability and mechanisms that modulate antibiotic affinity [7,10,11,12].

Murine models of sepsis and pneumonia caused by *A. baumannii* are thought to contribute to the understanding of the pathogenesis and the dynamics of the host response because these animals develop acute infections with pathological changes; they can therefore be used for the elaboration of new infection protocols and vaccine quality control and for the development of pharmacological interventions [13,14,15,16,17,18,19,20].

Most infection models use cyclophosphamide (CY) or virulence enhancement of *A. baumannii* (by mucin) that causes infections similar to those found in patients [21]. CY is the immunosuppressive drug most widely used in murine models to understand disease mechanisms and immune responses [21,22,23]. It is an immunosuppressive alkylating agent that causes a reduction in polymorphonuclear neutrophils; it predisposes the host to severe infections [24], inactivating the rapid cycle of the immune cell population, causing granulopenia [23,25,26] and decreasing B lymphocytes and T lymphocytes in the spleen and lymph nodes [26,27]. 

Vaccination strategies based on inactivated multi-drug-resistant bacterial strains mostly employ antigen adsorption, predominantly on aluminum hydroxide or Freund’s complete adjuvant [22,28,29,30,31,32,33]. On the other hand, new molecules such as chitosan have been used as vaccine adjuvants. Chitosan is a natural, non-toxic, biodegradable, and biocompatible polymer used in tissue engineering and drug release control that stimulates cellular immune responses and is more efficient and safer than incomplete Freund’s adjuvant or hydroxide aluminum [23,34].

Given this scenario, we propose a proof-of-concept study using murine models. We evaluated immune responses using CY-treated mice immunized with a vaccine containing inactivated whole cells of multi-drug-resistant *A. baumannii* adsorbed on aluminum hydroxide-chitosan (mAhC). We intend to contribute to the search for alternatives for protection against *A. baumannii* infections.

## 2. Materials and methods

### 2.1. A. baumannii Strain and Lethal Dose Determination

The *A. baumannii* strain was provided by the Central Laboratory of Tocantins (Lacen/TO) and stored at −80 °C in the Laboratory of Biomolecules and Vaccines of the Federal University of Tocantins (LaBVac/UFT). This strain was isolated in a hospital environment and characterized as a resistant carbapenem class for the blaOXA-23 and blaOXA-51 genes with oxacillinases performed by in-house multiplex polymerase chain reaction (Fiocruz/RJ). The strain was grown in 10 mL of tryptic soy broth (TSB) (soy peptone 17 g/L; casein 3 g/L; dextrose 5 g/L; potassium phosphate monobasic 5 g/L) under stirring at 120 rpm at 37 °C for 8 h. At the end of the cultivation, the entire volume was transferred to 90 mL of sterilized tryptic soy broth (TSB) and incubated under agitation at 120 rpm and 37 °C for 14 h. Subsequently, 30 mL of culture was centrifuged and washed in saline to standardize serial factor-4 dilutions (1/4–1/1024). Three mice were infected intraperitoneally with 500 µL of each of the above-described dilutions to determine the lethal dose and correlation with cell concentration (CFU/mL) [31].

### 2.2. Inactivation of A. baumannii Culture

*A. baumannii* cultures were incubated overnight in brain heart broth (Merck, Darmstadt, Germany) under agitation at 120 rpm at 37 °C. Cell densities were at the optical density: 600 nm (OD_600_) = 0.8. Subsequently, they were inactivated using 3% formaldehyde (Alphatec^®^, São Paulo, Brazil) for 2 h at 30 °C, followed by 16 h at 60 °C, under agitation at 120 rpm. Inactivation was confirmed by adding 1 mL of the inactivated culture to Mueller–Hinton agar (Kasvi^®^, Paraná, Brazil) and incubating at 37 °C for 48 h [32].

### 2.3. mAhC and Vaccine Formulation

The mAhC was prepared with prior chitosan solubilization using chitosan powder (Polymar, 81% deacetylation degree) in 0.8% (*v*/*v*) acetic acid, 0.9% (*w*/*v*) saline solution, with a ratio of chitosan and aluminum hydroxide of 2:1, respectively. Groups of four animals were employed to assess the number of mAhC doses. Female Swiss mice were inoculated intramuscularly with 100 µL of the mAhC (without *A. baumannii* antigen) in a single shot, at time zero (0 D), with double doses at times (0 D, 14 D) and triple doses at 0 D, 14 D, and 28 D. After the last inoculation, the animals were challenged with a lethal dose of intraperitoneally administered *A. baumannii* (500 µL), and were then monitored for clinical manifestations; then, the Kaplan–Meier survival curve was determined. The vaccine formulation consisted of a cell concentration of *A. baumannii* of 1 × 10^9^ cells.mL^−1^ at 1:1 (*v*/*v*) with chitosan matrix (1.25%, *w*/*w*) associated with aluminum hydroxide (1.6 mg) (Dinâmica^®^, Belo Horizonte, Brazil) [29].

### 2.4. mAhC and Molecular Docking Calculations

The ligand selected for the molecular docking study was chitosan, and its 3D structures were built in neutral form using Marvin Sketch 18.10, ChemAxon (http://www.chemaxon.com, accessed on 31 January 2023). *Acinetobacter baumannii* lipopolysaccharide (LPS) amino acid sequences were obtained from the UniProt server database (http://uniprot.org, accessed on 31 January 2023). The 3D structures of both proteins were constructed using the homology modeling approach with The Swiss Model Workspace (https://swissmodel.expasy.org/, accessed on 31 January 2023) after selecting their respective models using the BLASTp tool. The models were downloaded from The Protein Databank (https://www.rcsb.org/, accessed on 31 January 2023), considering quality parameters such as the experimental method, resolution, and R-value, and their complexation with a ligand. We used the Swiss model to verify protein structure breaks and amino acid positioning in the active site [35]. The generated models were validated by inspecting the Ramachandran plots [36,37], in which it was possible to analyze the distribution of the skeletal torsion angles φ and ψ responsible for the stereochemical quality of the protein studied, as well as the QMEAN factor [38]. Targets and ligands were prepared for the molecular docking process using Autodock Tools 1.5.7 [39], according to the methodology proposed by Moura et al. (2020) [40]. Using the AutoDock Vina [41] in the docking calculations, nine docking positions were generated for ligands interacting with the targets, returning affinity energy values (kcal/mol). The docking position results were analyzed using PyMOL 2.0 [42] and Discovery Studio 4.5 [43] to select the best position for each ligand inside the protein target using the parameters proposed by Moura et al. (2020) [40].

### 2.5. Ethical/Legal Requirements and Use of Animals in Experiments

This study used female Swiss mice aged 6 to 8 weeks weighing 17–23 g. They were acquired from the Central Animal Facility of the Federal University of Goiás and placed in the Laboratory of Biomolecules and Vaccines (LaBVac) pre-clinical testing room. The mice were housed in a pathogen-free primary enclosure with free access to food and water; the enclosure was maintained at 24 °C, with odor and light control with a light and dark cycle every 12 h. All animals in this experiment were previously dewormed with access to a 1:20 Ivermectin^®^ solution for seven days. The experiments were conducted following ethical recommendations established by the law of procedures for the scientific use of animals, being approved for execution by the Animal Ethics and Research Committee of UFT under protocol number 23101.002359/2020-31. Every effort was made to avoid suffering or undue pain; the mice were monitored for weight loss and gain and clinical manifestations such as lethargy, hypothermia, and difficulty breathing. The animals were euthanized using ketamine (300 mg/kg) (Vetbrands^®^, Rio de Janeiro, Brazil) and xylazine hydrochloride (22.5 mg/kg) (Syntec^®^, Piracicaba, Brazil) [17,18,19].

### 2.6. Experimental Design

Mice were given 3 intraperitoneal doses of CY (Genuxal^®^ Baxter, São Paulo, Brazil) (150 mg/Kg) for 7 days at 36 h intervals [29]. Blood samples were collected for hematologic analyses at the Pet Shop Dog Veterinary Clinic Center (Gurupi/TO-Brazil). Once the immunosuppression picture was established, 10 immunosuppressed Swiss mice were used to administer 100 µL of the vaccine composition in 3 doses at 0D, 14D, and 28D. Other groups of 10 immunosuppressed mice were inoculated with 100 µL of adjuvant solution (mAhC group) and phosphate-buffered saline (PBS) (non-immunized group) at the corresponding times of 0D, 14D, 28D. Then, 7 days after the final immunization (35D), mice were challenged with 500 μL of a previously defined and intraperitoneally administered lethal dose of *A. baumannii* [31]. The animals were monitored for clinical aspects, and biological samples were collected.

### 2.7. Enzyme-Linked Immunosorbent Assay Using A. baumannii Cultures

Blood samples from immunosuppressed mice after the last immunization (28D) and after the challenge test (35D) were evaluated to stimulate total IgG production. The ELISA was applied using *A. baumannii* culture, which used 10 mL of TSB for *A. baumannii*, under stirring at 120 rpm at 37 °C, for 8 h (pre-inoculum). Subsequently, the bacterial suspension was washed in saline solution, centrifuged to remove all the culture broth, and used for standardization and antigen fixation [33]. Cell concentration of 10^6^ CFU/mL was used for overnight sensitivity of 96-well microplates (Nunc MaxiSorp during) at 4 °C. After blocking for 1 h with 2% casein-PBS buffer (blocking buffer) at room temperature, the plates were washed 4 times with 0.05% Tween-20-PBS buffer. Serum was diluted at 1:50 with blocking buffer and incubated for 16 h at 4 °C. After washing, anti-mouse IgG peroxidase conjugate (Sigma, St. Louis, MO, USA) was used at a ratio of 1:2000 in the blocking buffer. After washing, 3,3′,5,5′-tetramethylbenzidine (BD Biosciences, San Diego, CA, USA) was added for 15 min and blocked with 2.5 M H_2_SO_4_ (Dinâmica^®^, Belo Horizonte, Brazil). Optical density was measured at 450 nm, and values above the cutoff were considered positive. The mean optical density of serum determined the cutoff point before immunization [17].

### 2.8. Histopathological Analysis

After finishing the lethal dose analysis test, the histopathological analysis of organs was performed by collecting the liver, lung, and spleen from three non-immunized, mAhC-treated, and immunized animals. The organs were weighed and fixed in a 10% formalin-buffered solution. Subsequently, they were dehydrated in alcohol (Qhemis^®^, São Paulo, Brazil), diaphanized in xylene (Synth^®^, Porto Alegre, Brazil), and embedded in histological paraffin (Synth^®^, Porto Alegre, Brazil). Then, 5-micron sections were made with a microtome, stained in hematoxylin/eosin, and examined under a Leica DM1000 LED optical microscope (Software Application Suite version 4.9.0) with 10, 40, and 100× objectives.

Histological slides of the liver, lung, and spleen were analyzed for tissue damage caused by *A. baumannii* infection, evaluating the presence of edema, degeneration, necrosis, acute inflammation, and hyperemia. Histopathological damage was classified using the lesion score considering the degree of alteration (if absent) when there was no evidence of damage, discrete if it was evident in up to 25% of the field of observation, moderate if greater than 25% but less than 50%, or severe if greater than 50% [17].

As previously described, expansion analyses of white pulp from the spleens of immunosuppressed mice were performed [44]. Histological sections of the spleen were analyzed by capturing images using a Leica Application Suite 4.9.0 photomicroscope. Histological images of white pulp were measured in 10 fields using a 10× objective and ImageJ^®^ 1.47v–2012 software (Bethesda, MD, USA) [44].

### 2.9. Statistical Analysis

Statistical analysis was performed using the GraphPad Prism software GraphPad Software Inc. San Diego, CA, USA. The Kruskal–Wallis analysis of variance test was used to compare non-immunized, mAhC-treated, and immunized animals (regarding immunization efficiency). The following parameters were compared: the number of surviving animals according to the Kaplan–Meier survival curves was used with the log-rank test (Mantel–Cox test) and those with symptoms, white blood cell count, and IgG production. Histopathological damage and white pulp expansion were used to compare non-immunized versus adjuvant-treated and non-immunized versus immunized animals. The significance level was *p* < 0.05, and we included the 95% confidence interval. The Student’s *t*-test was used for comparisons of means of experimental parameters.

## 3. Results

### 3.1. Sepsis Model of Infection Using A. baumannii in CY-Untreated Mice

The survival curves of mice after inoculation with an active culture of *A. baumannii* revealed percentages corresponding to 100%, 50%, and 0% when active suspensions were inoculated intraperitoneally at 10^−3^ (OD_600_ 0.02), 10^−2^ (OD_600_ 0.15), and 10^−1^ (OD_600_ 1.24), respectively, *p* < 0.01 (Figure 1A). After that, mice infections were performed using *A. baumannii* culture at 10^9^–10^7^ CFU/mL. After the first 24 h, *A. baumannii*-infected (4.0 × 10^8^ CFU/mL) mice showed 75% lethality, and this dose was used for the evaluations of vaccine efficacy (Figure 1B), *p* < 0.001.

### 3.2. Determination of mAhC Dose in CY-Untreated Mice

Mice (CY-untreated mice) inoculated with one or more doses of mAhC showed body-weight gain (Figure 2A). However, after inoculation with the lethal dose of *A. baumannii* (4.0 × 10^8^ CFU/mL), differences in survival percentages were observed, with the highest survival rate obtained when a higher number of doses was applied (Figure 2B). No survivors were observed after an *A. baumannii* challenge test in CY-untreated mice that received a single dose of mAhC or were administered only 1.6 mg/mL aluminum hydroxide solution. CY-untreated mice inoculated with 2 or 3 doses showed a survival rate of 45% (*p* < 0.01), suggesting the potential stimulation and protection of mAhC (Figure 2B).

### 3.3. Molecular Docking from MAhC and Bacterial Lipopolysaccharide

The selected templates for homology modeling are shown, highlighting the identities and the validation results with the corresponding Ramachandran favored values and QMEAN. (Table 1). The chitosan complexed with the receptor and formed various interactions with varying affinity energies, as indicated by the docking assays (Table 2). The chitosan ligand showed better affinity energy with target *Acinetobacter baumannii* at −8.4 kcal/mol. The complex formed between the chitosan ligand and the LPS target receptor of the organism *Acinetobacter baumannii* presents a strong interaction with the receptor’s amino acids (Figure 3A). This complex showed interactions between the target and ligand of the type: van der Waals, conventional hydrogen bond, carbon–hydrogen bond and pi-donor hydrogen bond (Figure 3B).

### 3.4. Immunosuppressive Model

We established an immunosuppressive model in Swiss mice using doses of CY, an agent with immunosuppressive properties that causes a decrease in hematologic parameters in the blood and leukopenia. CY promoted a progressive change in the clinical picture after three doses, causing ruffled hair, hair loss, and weight loss (*p* < 0.05) (Figure 4A). Furthermore, about 70% of the CY-treated mice survived and experienced a gradual recovery in body-weight gain (Figure 4B). In addition, these animals showed an exacerbated reduction in the number of total leukocytes, 79% (*p* < 0.001) (Figure 4C), neutrophils (*p* < 0.01) (Figure 4D), lymphocytes (*p* < 0.001) (Figure 4E), and monocytes (*p* < 0.05) (Figure 4F), compared to the group of non-treated mice. Erythrocytes reduction was also evidenced with a decline of 51% compared to the group of CY-treated mice (*p* < 0.01) (Figure 4G) and a drop in hemoglobin concentration (*p* < 0.05) (Figure 4H) and hematocrit (*p* < 0.05) (Figure 4I).

### 3.5. Protective Effect of A. baumannii Whole-Cell Vaccine Inactivated and Adsorbed on mAhC in CY-Treated Mice

Analysis of the post-immunization humoral response revealed greater stimulation of IgG levels in CY-treated mice inoculated with mAhC alone (adjuvant group) compared to non-immunized CY-treated mice (*p* < 0.05) (Figure 5). However, CY-treated mice immunized with inactivated *A. baumannii* whole-cell vaccine adsorbed with mAhC had IgG levels significantly higher than the levels obtained from CY-treated mice in the adjuvant group (*p* < 0.05), suggesting a greater capacity to stimulate the immune response against *A. baumannii* (Figure 5).

The study of vaccine efficacy after applying the lethal dose of active *A. baumannii* (4 × 10^8^ CFU/mL) revealed a lethality rate of 100%, corresponding to non-immunized and CY-treated mice (control group) (*p* < 0.001). In contrast, in immunized CY-treated mice, the survival rate was 85%, a value that differed from the survival percentages of CY-treated mice in the mAhC group (adjuvant) (45%) (*p* < 0.05) (Figure 6A). In addition, immunized CY-treated mice showed higher levels of IgG compared to the control and adjuvant groups (Figure 6B) (*p* < 0.05), as well as a more significant expansion of white pulp (Figure 6C,D).

### 3.6. Histopathological Analysis of the Sepsis Model of A. baumannii Infection in CY-Treated Mice

The sepsis model of *A. baumannii* infection showed tissue damage in the liver, spleens, and lungs in CY-treated mice, the non-immunized mice, and the mAhC (Figure 7). By comparison, there were no significant differences between the liver and lung weights of these groups (Figure 7A,B). In terms of the weight of the spleens of immunized CY-treated mice, there were significant differences from the non-immunized CY-treated or adjuvant groups (*p* < 0.05) (Figure 7C). In addition, more substantial expansion of the white pulp was observed in this group (*p* < 0.01) and the adjuvant group (*p* < 0.05) (Figure 7D). The expansion of the white pulp is related to the presence of periarteriolar and follicular hyperplasia caused by the antigenic stimulus and immune response (Figure 6D and Figure 7D). The histopathological findings revealed greater severity in the liver with hydropic degeneration and decreased sinusoids, and interstitial pneumonia with a mixed inflammatory infiltrate and congestion (of minor severity in the mAhC and immunized groups) in the lung. Additionally, hemorrhage, congestion, and white pulp lymphoid hyperplasia were detected in the spleen (Figure 7D).

## 4. Discussion

*A. baumannii* is a critical emerging pathogen that is responsible for nosocomial pneumonia in healthcare settings; it has become a global emergency [45,46,47]. Murine models have contributed to elucidating the mechanisms associated with pathogenesis and understanding the pathogen–host relationship of infections caused by *A. baumannii* in sepsis, pneumonia, and immunosuppression models [17,19,24]. CY is a synthetic antineoplastic agent widely used as an immunosuppressive drug in mice, which causes a cytostatic effect and inhibits the rapid cycle of immune cells, consequently causing granulopenia [23,25,26]. *A. baumannii*-infected mice may show changes consistent with pneumonia, acute inflammation with mild-to-severe inflammatory infiltration of polymorphonuclear cells, segmental abscess formation, and mild-to-moderate infiltration of alveolar macrophages when treated with an immunosuppressant [48]. Here, we used an *A. baumannii* sepsis model in CY-treated mice to understand the behavior of multi-drug-resistant strains in infections and to evaluate new vaccine strategies based on inactivated bacterial strains adsorbed on mAhC.

Preliminarily, we revealed the immunosuppression of immunocompetent mice that showed signs of infection and presented with ruffled and shedding animal hair, loss in body weight, and lethargy, accompanied by a reduction in leukocytes (neutrophils, lymphocytes, and monocytes), erythrocytes, and hemoglobin and the percentage of hematocrit (Figure 4). Our data corroborate those of other studies that show low concentrations of circulating polymorphonuclear neutrophils, followed by a marked and progressive increase leading to greater susceptibility of immunosuppressed mice [23,29,49] and distinct alterations of macrophages and lymphocytes with lower participation in the first hours of infection, followed by baseline levels.

The expansion of the follicular area is characterized by increased cellularity of rounded and regular nuclear membranes stimulated by immunization, leading to subsequent B cell proliferation [17,28,50,51]. The present study revealed the expansion of spleen white pulp in the immunized CY-treated mice groups with the production of IgG antibodies. In addition, the adequate protection of 85% after a lethal dose of multi-drug-resistant *A. baumannii* suggests that our murine model was effective in vaccine protection in immunized animals (Figure 6B,C), as reported in previous studies [23,28,30].

Another relevant aspect observed here was the greater tissue protection of groups of immunized CY-treated mice because non-immunized CY-treated mice or adjuvant-inoculated-CY-treated mice presented histopathological findings of edema, hemorrhage, hydropic degeneration, necrosis, and hyperemia in liver sections, as well as edema, hemorrhage, and hyperemia in lung sections. There were signs of acute infection in the non-immunized CY-treated mice and adjuvant-inoculated-CY-treated mice groups after infection, revealing characteristic signs of the pathology with death within 24 h. In contrast, the immunized CY-treated mice groups showed adequate protection after receiving a lethal dose of multi-drug-resistant *A. baumannii*. Our data corroborate previous infection studies in murine sepsis models [49,52,53].

Splenic alterations caused by *A. baumannii* infections are accompanied by white pulp damage, leukocyte apoptosis, embolus formation, clusters of degenerated or necrotic leukocytes in the white pulp areas, and the appearance of moderate congestion in the medullary region [49,54]. Infiltration of mixed inflammatory cells in the perivascular and peribronchial space, with the presence of polymorphonuclear granulocytes and mononuclear cells in the airway lumen, especially in the first hours of infection, is followed by vasodilation, congestion, and hemorrhage, which leads to the destruction of alveolar structures of the lung after 48 h of infection [49,52,53,55,56]. Our histopathological findings highlight the importance and adequacy of the sepsis model, evidencing vaccine protection, the stimulation of the immune response, and the consequent tissue protection of the livers, spleens, and lungs of immunized animals.

The mAhC used here contributed to the stimulation of IgG production of immunized CY-treated mice groups and was superior to non-immunized CY-treated mice groups when used at three doses/animal. Previous studies demonstrated the stimulation of IgG production in immunized mice using protein antigens adsorbed exclusively on chitosan [34,57,58] or exclusively on aluminum hydroxide or Freund’s adjuvant [15,30,33,59]. However, the present study is the first to associate aluminum hydroxide and chitosan for the adsorption of *A. baumannii* antigens on mAhC, revealing a vaccine adjuvant that stimulates the immune system response of immunosuppressed mice.

Despite the relevance of the efficacy of a new vaccine formulation with inactivated multi-drug-resistant *A. baumannii* that is whole-cell adsorbed on mAhC, issues remain that have not been fully elucidated. As described here, we recorded high stimulation of the immune response by measuring total IgG levels. However, we did not determine IgG1 and IgG2c levels, which participate in eliminating pathogens and reducing postoperative bacterial loads [17,32,33]. We did not measure cytokine profiles, which may be necessary for elucidating the mechanism of pathogen elimination and vaccine protection. Thus, future studies should address these issues.

## 5. Conclusions

We demonstrated the role of an *A. baumannii* sepsis model in mice and the reduction in neutrophil numbers before immunization (treated with CY). The inactivated and adsorbed multi-drug *A. baumannii* whole-cell vaccine (mAhC) stimulated the immune response and protected immunized animals, even after a challenge with a lethal dose. These findings provide the basis for studies on minimizing the impacts of *A. baumannii* infections.

## Figures and Tables

**Figure 1 vaccines-11-00669-f001:**
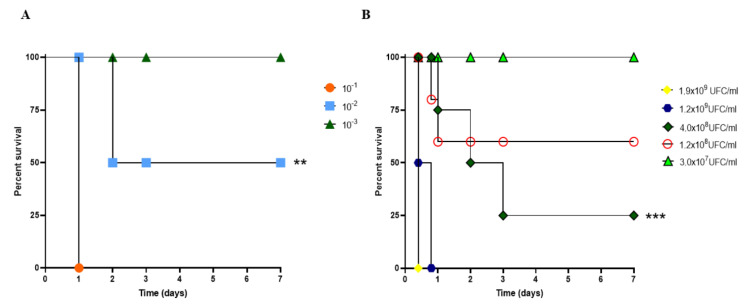
Determination of the lethal dose of *A. baumannii* in CY-untreated mice. (**A**) The survival rate of mice subjected to bacterial challenge using serial dilution ranged from 10^−1^ to 10^−3^. (**) *p* < 0.01). (**B**) Survival curve of mice submitted to bacterial challenge using colony-forming unit (CFU/mL of *A. baumannii*). *p*-values determined by Kaplan–Meier survival curves were used with the log-rank test (Mantel–Cox test, with six animals per group). (***) *p* < 0.001. Immunized CY-treated mice.

**Figure 2 vaccines-11-00669-f002:**
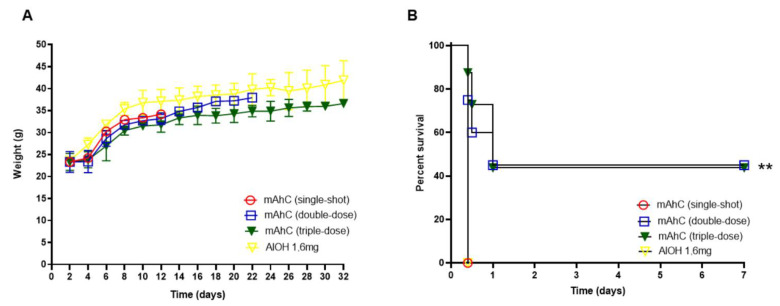
Determination of the number of mAhC doses in CY-untreated mice. (**A**) Bodyweight monitoring of CY-untreated mice. Data are expressed as the mean and standard deviation. *p* < 0.05). (**B**) Survival curve of mice subjected to *A. baumannii* challenge (4.0 × 10^8^ CFU/mL). The log-rank test (Mantel–Cox test, with six animals per group) determined the *p*-value. (**) *p* < 0.01.

**Figure 3 vaccines-11-00669-f003:**
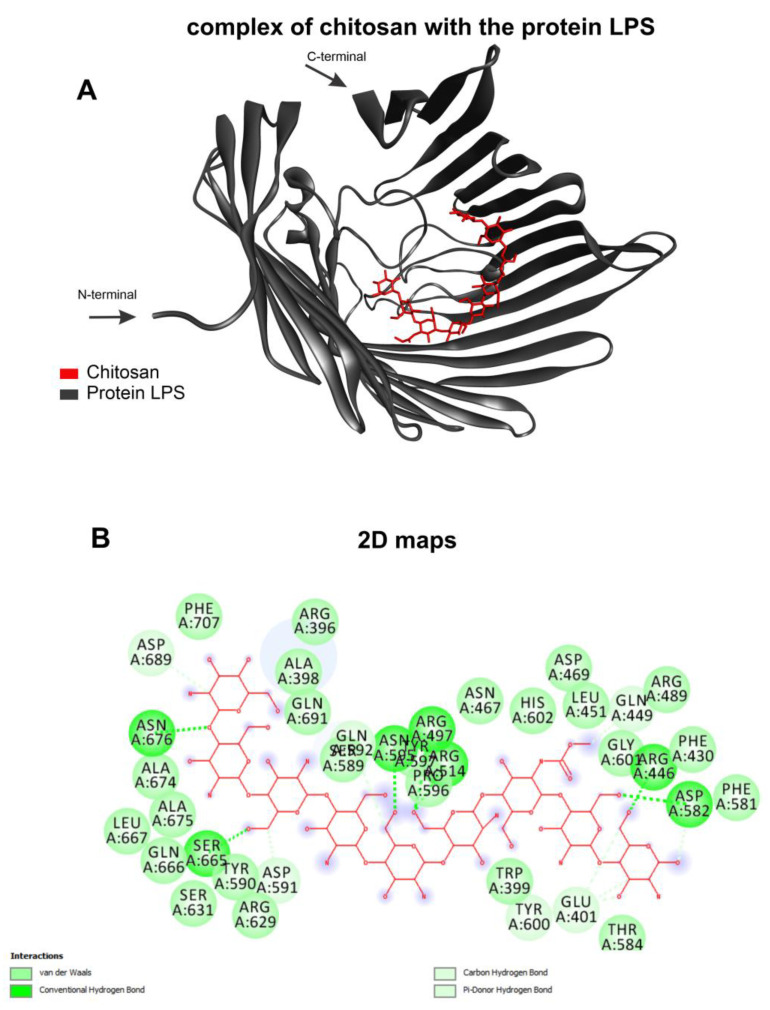
Molecular docking calculations. Chitosan (red) complexed with protein LPS (gray) (**A**) and 2D maps of molecular interactions with amino acids (**B**) of *Acinetobacter baumannii*.

**Figure 4 vaccines-11-00669-f004:**
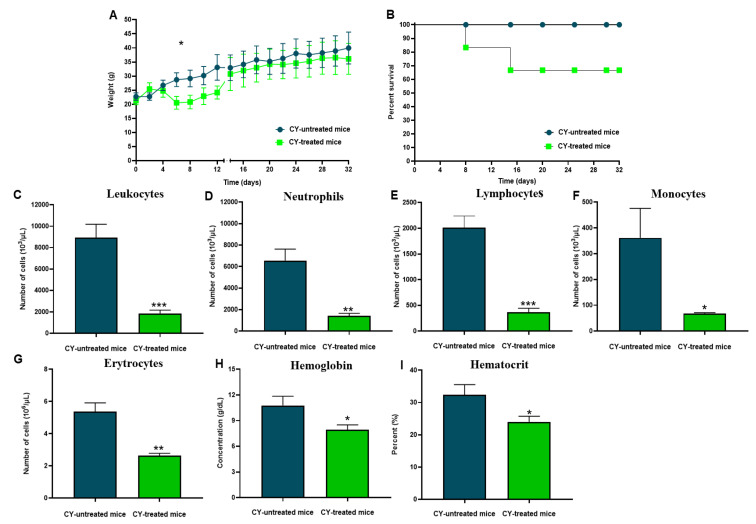
Description of the immunosuppressive model Swiss mice. (**A**) Bodyweight monitoring of CY-treated and CY- untreated mice using six animals per group. Data are expressed as the mean and standard deviation, (*) *p* < 0.05. (**B**) Percentage survival of CY-treated mice using cyclophosphamide (150 mg/kg). (**C**) Leukocyte numbers. (***) *p* < 0.001. (**D**) The number of neutrophils. (**) *p* < 0.01. (**E**) Several lymphocytes. (***) *p* < 0.001. (**F**) The number of monocytes. (*) *p* < 0.05. (**G**) Several erythrocytes. (**) *p* < 0.01. (**H**) Hemoglobin. (*) *p* < 0.05. (**I**) Hematocrit. (*) *p* < 0.05. *p* values were determined by the *t*-test.

**Figure 5 vaccines-11-00669-f005:**
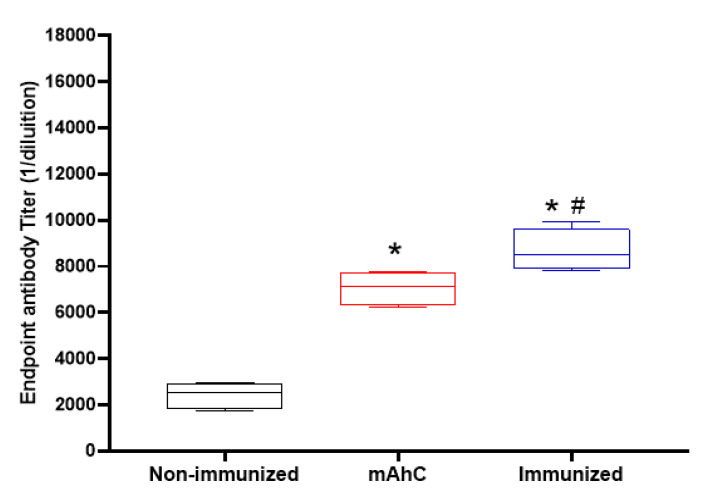
Stimulation of the humoral response of inactivated *A. baumannii* whole-cell vaccine adsorbed on mAhC in CY-treated mice. Data refer to the levels of total IgG produced before the application of a lethal dose of *A. baumannii* (4.0 × 10^8^ CFU/mL) in immunized CY-treated mice, non-immunized CY-treated mice, and CY-treated mice inoculated with mAhC using 6 animals per group. (*) *p* < 0.05 represents a comparative analysis between non-immunized CY-treated mice vs. CY-treated mice inoculated with mAhC using 6 animals per group. (*#) *p* < 0.05 represents a comparative analysis between immunized CY-treated mice vs. CY-treated mice inoculated with mAhC.

**Figure 6 vaccines-11-00669-f006:**
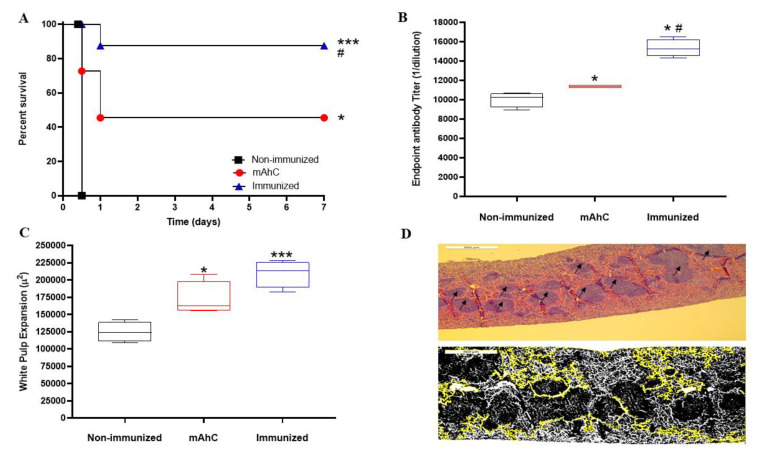
Efficiency test of whole-cell vaccine inactivated and adsorbed on mAhC in CY-treated mice. (**A**) Survival curve of experimental groups using CY-treated mice after a lethal dose of *A. baumannii* (4.0 × 10^8^ CFU/mL). The log-rank test (Mantel–Cox test, with six animals per group) determined the *p*-value. (*) *p* < 0.05 represents a comparative analysis between non-immunized CY-treated mice vs. CY-treated mice inoculated with mAhC using six animals per group. (***) *p* < 0.001 represents a comparative analysis between non-immunized CY-treated mice vs. immunized CY-treated mice and (#) *p* < 0.05 represents a comparative analysis between immunized CY-treated mice vs. CY-treated mice inoculated with mAhC. (**B**) Humoral response after a lethal dose of *A. baumannii* (4.0 × 10^8^ CFU/mL). Data refer to the total IgG levels produced from immunized CY-treated mice, non-immunized CY-treated mice, and post-lethal CY-treated mice inoculated with mAhC. (*) *p* < 0.05 represents a comparative analysis between non-immunized CY-treated mice vs. CY-treated mice inoculated with mAhC and non-immunized CY-treated mice vs. immunized CY-treated mice. *p* < 0.05 (*#) represents a comparative analysis between immunized CY-treated mice vs. CY-treated mice inoculated with mAhC. (**C**,**D**) Measurement and imaging of white pulp expansion of immunosuppressed mice. (*) *p* < 0.05 represents a comparative analysis between non-immunized CY-treated mice vs. those inoculated with mAhC. (***) *p* < 0.01 represents a comparative analysis between immunized CY-treated mice vs. CY-treated mice inoculated with mAhC. ImageJ employee software is available (https://imagej.nih.gov/ij/ accessed on 31 January 2023).

**Figure 7 vaccines-11-00669-f007:**
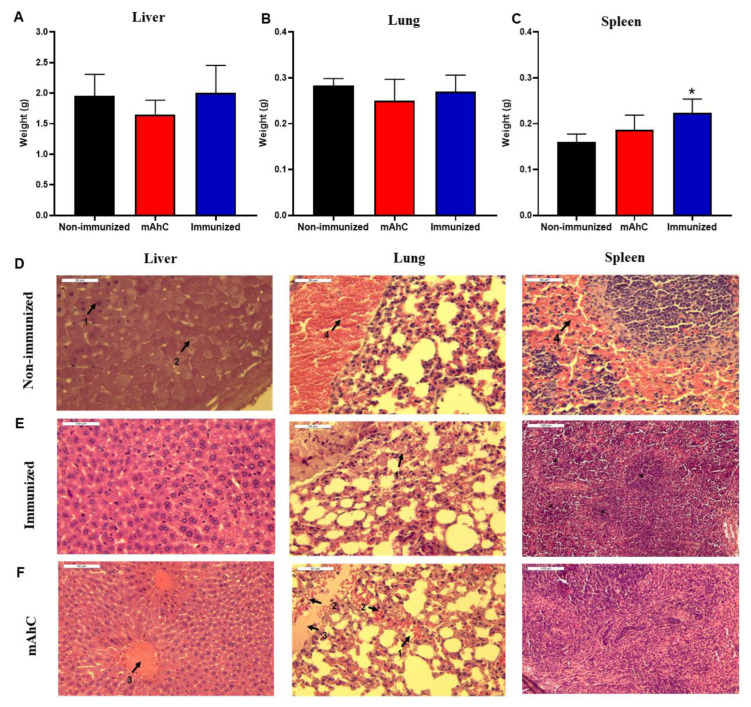
Histopathological findings of *A. baumannii* infection in CY-treated mice. (**A**–**C**) Determination of the lung, liver, and spleen weights of CY-treated mice after the application of a lethal dose of *A. baumannii* (4.0 × 10^8^ CFU/mL) using 6 animals per group. Data are expressed as the mean and standard deviation. (*) *p* < 0.05 represents a comparative analysis between non-immunized CY-treated mice vs. immunized CY-treated mice. (**D**–**F**) Histopathological findings of experimental groups using CY-treated mice after the application of a lethal dose of *A. baumannii* (4.0 × 10^8^ CFU/mL) stained by the hematoxylin-eosin technique. Data: D1: necrosis (40×), D2: hydropic Degeneration (40×), D3: hyperemia (40×), D4: hemorrhage (40×), E1: hemorrhage (40×), F1: pulmonary hemorrhage (40×), F2: polymorphonuclear neutrophils (40×), F3: acute inflammation (40×), F4: hyperemia (40×), F* white pulp.

**Table 1 vaccines-11-00669-t001:** The target model of the *A. baumannii* lipopolysaccharide was used to analyze the molecular docking with chitosan.

Organism	Target (Unipot Database)	Template	Identity (%)	Ramachandran Favored (%)	QMEAN
*A. baumannii*	LPS (D0C7T1)	5IV8	34.02%	90.54%	0.56

**Table 2 vaccines-11-00669-t002:** Molecular docking results for complexes between chitosan and the target of *A. baumannii*.

Organism	Ligand	Affinity Energy (kcal/mol)
*A. baumannii*	Chitosan	−8.4

## Data Availability

No new data were created or analyzed in this study. Data sharing is not applicable to this article.

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
