# Peer review of "Using an Aluminum Hydroxide–Chitosan Matrix Increased the Vaccine Potential and Immune Response of Mice against Multi-Drug-Resistant Acinetobacter baumannii"

_vaccines, 2023, doi:10.3390/vaccines11030669_

Round 1
Reviewer 1 Report
Deusdara et al. characterized in the present manuscript that the aluminum hydroxide-chitosan matrix increased mice's vaccine potential and immune response of mice againts multidrug-resistant Acinetobacter baumannii. They evaluated a multidrug-resistant A. baumannii whole-cell vaccine, inactivated and adsorbed on an aluminum hydroxide-chitosan (mAhC) matrix, in an A. baumannii sepsis model in immunosuppressed mice. The manuscript of Deusdara and colleagues points towards an important and somewhat underestimated aspect of protection against A. baumannii infections. Unfortunately, the manuscript lacks the necessary controls and should be significantly revised and extended before it is suitable for publication in this journal. Specifically, the following points should be addressed:
- The ability to grow in a complex medium such as TSB broth is not an appropriate control to test for growth defects that might occur in vivo. Please use a suitable minimal medium (e.g., M9) and/or a medium mimicking the intracellular environment A. baumannii is exposed to within host cells. Also, see the growth of A. baumannii with and without chitosan and aluminum hydroxide. It would be highly desirable.
- The author must explain what kind of immune repose is better for eliminating A. baumannii.
- What are the status of secretary IgA and CD4 and CD8 ration?
- Line no. 224 p small in the case and italic (see all places)
- Figure 3 A labels the n and c terminals side by an arrow.
- The authors should explain in Figure 7 that no much difference is observed in the liver and lungs.
- The manuscript should be carefully proofread in the English language.
Author Response
Reviewer's comments:
Point 1. Deusdara et al. characterized in the present manuscript that the aluminum hydroxide-chitosan matrix increased mice's vaccine potential and immune response of mice againts multidrug resistant Acinetobacter baumannii. They evaluated a multidrug-resistant A. baumannii whole cell vaccine, inactivated and adsorbed on an aluminum hydroxide-chitosan (mAhC) matrix, in an A. baumannii sepsis model in immunosuppressed mice. The manuscript of Deusdara and colleagues points towards an important and somewhat underestimated aspect of protection against A. baumannii infections.
Reply: We are grateful for the opportunity to revise our manuscript, and we thank the review for investing the time and effort to increase the quality of our work. All concerns/comments were addressed in the present version of the manuscript. Please refer the text below.
Point 2. Unfortunately, the manuscript lacks the necessary controls and should be significantly revised and extended before it is suitable for publication in this journal. Specifically, the following points should be addressed: The ability to grow in a complex medium such as TSB broth is not an appropriate control to test for growth defects that might occur in vivo. Please use a suitable minimal medium (e.g., M9) and/or a medium mimicking the intracellular environment A. baumannii is exposed to within host cells.
Reply: Thank you for questions. We would like to emphasize that the present manuscript was focused on the use of the aluminum hydroxide-chitosan matrix as a vaccine adjuvant. As described on 69 – 72, we highlight the fact that chitosan is a natural, non-toxic, biodegradable, and biocompatible polymer used in tissue engineering and drug release control that stimulates cellular immune responses and is more efficient and safer than incomplete Freund's adjuvant or hydroxide aluminum (Dollery et. al., 2021; Sari et. Al., 2016). Therefore, we used TSB broth only for antigen production (A. baumannii). TSB, Muller-Hinton, BHI are specific culture media for A. baumannii (Castilho et al., 2017). The strain employed was isolated by the Central Reference Laboratory of Bacterial Isolates of the Tocantins State/Brazil and characterized by the Oswaldo Cruz Foundation/Brazil.
Reference:
Castilho, S. R. A.; Godoy, C. S. D. M.; Guilarde, A. O.; Cardoso, J. L.; André, M. C. P.; Junqueira-Kipnis, A. P.; Kipnis, A. Acinetobacter baumannii strains isolated from patients in intensive care units in Goiania, Brazil: Molecular and drug susceptibility profiles. PLoS One, 2017, 12, e0176790. https://doi.org/10.1371/journal.pone.0176790.
Dollery, S.J.; Zurawski, D.V.; Gaidamakova, E.K.; Matrosova, V.Y.; Tobin, J.K.; Wiggins, T.J.; Tobin, G. J. Radiation-Inactivated Acinetobacter baumannii Vaccine Candidates. Vaccines, 2021, 9, 96.
Sari, R. S.; Almeida, A. C.; Cangussu, A. S. R.; Jorge, E. V.; Mozzer, D. O.; Santos, H. O.; Quintilio, W.; Brandi, I. V.; Andrade, V. A.; Miguel, A. S. M.; Santos, E. M. S. Anti-botulism single-shot vaccine using chitosan for protein encapsulation by simple coacervation. Anaerobe, 2016, 42, 182-187. https://doi.org/10.1016/j.anaerobe.2016.10.013
Point 3. Also, see the growth of A. baumannii with and without chitosan and aluminum hydroxide. It would be highly desirable.
Reply: Again, we would like to emphasize that the present manuscript was focused on the use of the aluminum hydroxide-chitosan matrix (mAhC) as a vaccine adjuvant. TSB, Muller-Hinton, BHI are specific culture media for A. baumannii. In the present work, we used the inactivated antigen adsorbed on mAhC (A. baumannii) and also only mAhC. See Figure 2.
Reference:
Castilho, S. R. A.; Godoy, C. S. D. M.; Guilarde, A. O.; Cardoso, J. L.; André, M. C. P.; Junqueira-Kipnis, A. P.; Kipnis, A. Acinetobacter baumannii strains isolated from patients in intensive care units in Goiania, Brazil: Molecular and drug susceptibility profiles. PLoS One, 2017, 12, e0176790. https://doi.org/10.1371/journal.pone.0176790.
Dollery, S.J.; Zurawski, D.V.; Gaidamakova, E.K.; Matrosova, V.Y.; Tobin, J.K.; Wiggins, T.J.; Tobin, G. J. Radiation-Inactivated Acinetobacter baumannii Vaccine Candidates. Vaccines, 2021, 9, 96.
Sari, R. S.; Almeida, A. C.; Cangussu, A. S. R.; Jorge, E. V.; Mozzer, D. O.; Santos, H. O.; Quintilio, W.; Brandi, I. V.; Andrade, V. A.; Miguel, A. S. M.; Santos, E. M. S. Anti-botulism single-shot vaccine using chitosan for protein encapsulation by simple coacervation. Anaerobe, 2016, 42, 182-187. https://doi.org/10.1016/j.anaerobe.2016.10.013
Point 4. The author must explain what kind of immune repose is better for eliminating A. baumannii. What are the status of secretary IgA and CD4 and CD8 ration?
Reply: Thank you for your questions. We recorded high stimulation of the immune response by measuring total IgG levels and Kaplan-Meier survival curves (Figure 6). As described on line 404, we highlight the fact that, despite the relevance of the efficacy of a new vaccine formulation with inactivated multidrug-resistant A. baumannii whole-cell adsorbed on mAhC, issues remain that have not been fully elucidated. However, we did not determine IgG1 and IgG2c levels, which participate in eliminating pathogens and reducing postoperative bacterial loads (Ramezanalizadeh et al., 2020; Cangussu et. al., 2018; Shu et al. 2016) . We did not measure cytokine profiles, which may be necessary for eluci-dating the mechanism of pathogen elimination and vaccine protection. Future studies should address these issues.
Reference:
Ramezanalizadeh, F.; Owlia, P.; Rasooli, I. Type I pili, CsuA/B and FimA induce a protective immune response against Acinetobacter baumannii. Vaccine, 2020, 38, 5436-5446.
Cangussu, A.S.R.; Mariúba, L.A.M.; Lalwani, P.; Pereira, K.D.; Astolphi-Filho, E.S.; Orlandi, P.P.; Nogueira, P. A. A hybrid protein containing MSP1a repeats and Omp7, Omp8 and Omp9 epitopes protect immunized BALB/c mice against anaplasmosis. Vet. Res., 2018, 49, 1-11.
Shu, M.H.; MatRahim, N.; NorAmdan, N.; Pang, S.P.; Hashim S.H.; Phoon, W.H.; AbuBakar, S. An inactivated antibiotic-exposed whole-cell vaccine enhances bactericidal activities against multidrug-resistant Acinetobacter baumannii. Sci. Rep., 2016, 6, 1-8. https://doi.org/10.1038/srep22332
Point 5. Line no. 224 p small in the case and italic (see all places)
Reply: Yes. We rewrote this sentence to avoid misunderstanding.
Point 6. Figure 3 A labels the n and c terminals side by an arrow.
Reply: Yes! Edited as requested!
Point 7. The authors should explain in figure 7 that no much difference is observed in the liver and lungs.
Reply: Yes. We rewrote this sentence and figure to avoid misunderstanding.
Point 8. The manuscript should be carefully proofread in the English language.
Reply: In the current version, we addressed all comments/suggestions raised by the reviewers and modified the text accordingly. We also submitted the manuscript to be an English language specialist who reviewed the grammar and provided the adjustments. Attached is the certificate of Liberty Medical Communications, LLC. Philip Lindeman, MD, PhD, Boston, MA, USA. Attached, please find Certificate of Language Editing.
Reviewer 2 Report
As the title reflects, the manuscript describes a study of the potential of an inactivated vaccine absorbed in aluminum hydroxide-chitosan matrix in the generation of a good immune response against a cabapenem-resistant strain of A. baumannii. Overall, is a clear and concise manuscript, with an appropriate methodology. In short, this is a quality study with interesting implications for the development of alternatives in the prevention of infections against Acinetobacter baumannii, a bacterium considered by WHO as Priority 1. However, some recommendations, as well as some clarifications, should be taken into account prior to its publication. Once these reconmendations and clarifications are addressed, the article should be published.
1) The text quality of all figures should be improved so that they can be read clearly.
2) In the introduction, the use of murine models for sepsis and pneumonia caused by A. baumannii (lines 56 to 60) is referenced with 8 references, most of which are self-citations, which is excessive.
3) In Figure 6D, the arrows indicate the areas of white pulp expansion?
4) In the discusion, the authors state that "Another relevant aspect observed here was the greater tissue protection of groups of immunized CY-treated mice because non-immunized CY-treated mice or adjuvant-inoculated-CY-treated mice presented histopathological findings of edema, hemorrhage, hydropic degeneration, necrosis, and hyperemia in liver sections, as well as edema, hemorrhage, and hyperemia in lung sections." (lines 375 to 379). However, in Figure 7 the only histological finding shown in adjuvant-inoculated-CY-treated mice is hemorrhage in the lungs (E1), while in immunized CY-treated mice different effects are marked, both in the liver (Hyperemia F4) and in the lungs (F1, F2, F3). It is possible that the histologic sections are misidentified in Figure 7? I consider that this should be clarified especially for those readers unfamiliar with the images of these tissues.
Author Response
Reviewer's comments:
Point 1. As the title reflects, the manuscript describes a study of the potential of an inactivated vaccine absorbed in aluminum hydroxide-chitosan matrix in the generation of a good immune response against a cabapenem-resistant strain of A. baumannii. Overall, is a clear and concise manuscript, with an appropriate methodology. In short, this is a quality study with interesting implications for the development of alternatives in the prevention of infections against Acinetobacter baumannii, a bacterium considered by WHO as Priority 1. However, some recommendations, as well as some clarifications, should be taken into account prior to its publication. Once these reconmendations and clarifications are addressed, the article should be published.
Reply: We are grateful for having the opportunity to revise our manuscript, and we thank the reviewer for investing the time and effort to increase the quality of our work. All concerns/comments were addressed in the present version of the manuscript. Please, see the text below. We further thank for recognition of our efforts.
Point 2. The text quality of all figures should be improved so that they can be read clearly.
Reply: Yes! Edited as requested!
Point 3. In the introduction, the use of murine models for sepsis and pneumonia caused by A. baumannii (lines 56 to 60) is referenced with 8 references, most of which are self-citations, which is excessive.
Reply: Thank you for questions. We believe that the inserted references are relevant and complementary to the use use of murine models and applications of techniques such as bioinformatics.
Point 4. In Figure 6D, the arrows indicate the areas of white pulp expansion?
Reply: Yes!
Point 5. In the discusion, the authors state that "Another relevant aspect observed here was the greater tissue protection of groups of immunized CY-treated mice because non-immunized CYtreated mice or adjuvant-inoculated-CY-treated mice presented histopathological findings of edema, hemorrhage, hydropic degeneration, necrosis, and hyperemia in liver sections, as well as edema, hemorrhage, and hyperemia in lung sections." (lines 375 to 379). However, in figure 7 the only histological finding shown in adjuvant-inoculated-CY-treated mice is hemorrhage in the lungs (E1), while in immunized CY-treated mice different effects are marked, both in the liver (Hyperemia F4) and in the lungs (F1, F2, F3). It is possible that the histologic sections are misidentified in Figure 7? I consider that this should be clarified especially for those readers unfamiliar with the images of these tissues.
Reply: Yes! Done as requested by reviewer 1! We rewrote this sentence and figure to avoid misunderstanding.
Round 2
Reviewer 1 Report
Authors should understand vaccines are essential tools for preventing and controlling infectious diseases. The mechanisms of action for adjuvant are one vaccine formulation and challenge in the research and development of vaccines. Several criteria should be seen for developing adjuvants for vaccines: effectiveness in the target species, long-lasting protective immunity, safety, and last but not least, cost-effectiveness.
Immunoactive Compounds (Cytokines, IgG, IgE & IgM antibodies) are critical players in immune responses. These parameters are needs evaluate experimentally to convince your findings.
Author Response
Reviewers' comments:
Reviewer 1 – Round 2
Point – 1. Authors should understand vaccines are essential tools for preventing and controlling infectious diseases. The mechanisms of action for adjuvant are one vaccine formulation and challenge in the research and development of vaccines. Several criteria should be seen for developing adjuvants for vaccines: effectiveness in the target species, long-lasting protective immunity, safety, and last but not least, cost-effectiveness. Immunoactive Compounds (Cytokines, IgG, IgE & IgM antibodies) are critical players in immune responses. These parameters are needs evaluate experimentally to convince your findings.
Reply: We thank the reviewer for the time and efforts invested to reviewer our manuscript. The comments/concerns regarding the vaccines’ safety are very relevant and need to be addressed in the understanding of potential effects of novel molecules. For instance, our research group has invested time and efforts to assess the effects of cytokines and subclases of immunoglobulins that are related to A. baumanni infections. We understand that our experimental model represents a relevant step in such direction, allowing considerable advances, not only showing novel results regarding the use of chitosan as an adjuvant molecule, but also demonstrating the chitosan-mediated contributions for the the immune response of immunized hosts. Our ongoing efforts are focusing on the conduction of immunoactive compounds determinations (Cytokines, IgG, IgE & IgM antibodies), as correctly pointed out by the reviewer but beyond the scope of the current manuscript, for a full comprehension of the criteria involved in the development of a vaccine.